# Do female sex workers have lower uptake of HIV treatment services than non-sex workers? A cross-sectional study from east Zimbabwe

Rebecca Rhead,[1] Jocelyn Elmes,[1] Eloghene Otobo,[1] Kundai Nhongo,[2] Albert Takaruza,[2] Peter J White,[1,3,4] Constance Anesu Nyamukapa,[1,2] Simon Gregson[1]

RR and JE contributed equally.

[1]Department of Infectious Disease Epidemiology, Imperial College London School of Public Health, London, UK
[2]Biomedical Research and Training Institute, Harare, Zimbabwe
[3]MRC Centre for Outbreak Analysis and Modelling and NIHR Health Protection Research Unit in Modelling Methodology, Imperial College London School of Public Health, London, UK
[4]Modelling and Economics Unit, National Infection Service, Public Health England, London, UK

**Correspondence to**
Dr Rebecca Rhead;
r.rhead@imperial.ac.uk

## ABSTRACT

**Objective** Globally, HIV disproportionately affects female sex workers (FSWs) yet HIV treatment coverage is suboptimal. To improve uptake of HIV services by FSWs, it is important to identify potential inequalities in access and use of care and their determinants. Our aim is to investigate HIV treatment cascades for FSWs and non-sex workers (NSWs) in Manicaland province, Zimbabwe, and to examine the socio-demographic characteristics and intermediate determinants that might explain differences in service uptake.

**Methods** Data from a household survey conducted in 2009–2011 and a parallel snowball sample survey of FSWs were matched using probability methods to reduce under-reporting of FSWs. HIV treatment cascades were constructed and compared for FSWs (n=174) and NSWs (n=2555). Determinants of service uptake were identified a priori in a theoretical framework and tested using logistic regression.

**Results** HIV prevalence was higher in FSWs than in NSWs (52.6% vs 19.8%; age-adjusted OR (AOR) 4.0; 95% CI 2.9 to 5.5). In HIV-positive women, FSWs were more likely to have been diagnosed (58.2% vs 42.6%; AOR 1.62; 1.02–2.59) and HIV-diagnosed FSWs were more likely to initiate ART (84.9% vs 64.0%; AOR 2.33; 1.03–5.28). No difference was found for antiretroviral treatment (ART) adherence (91.1% vs 90.5%; P=0.9). FSWs' greater uptake of HIV treatment services became non-significant after adjusting for intermediate factors including HIV knowledge and risk perception, travel time to services, physical and mental health, and recent pregnancy.

**Conclusion** FSWs are more likely to take up testing and treatment services and were closer to achieving optimal outcomes along the cascade compared with NSWs. However, ART coverage was low in all women at the time of the survey. FSWs' need for, knowledge of and proximity to HIV testing and treatment facilities appear to increase uptake.

## Strengths and limitations of this study

► We provide novel insight into the differential uptake of HIV treatment services for female sex workers (FSWs) and non-sex workers (NSWs) in Manicaland province, Zimbabwe, and the personal, social and structural factors associated with these inequalities.

► We use data taken from a Manicaland household survey and a parallel snowball sample survey of FSWs, thus drawing on the strengths of population surveys and targeted approaches for hard-to-reach populations.

► Our study is unique in that it compares uptake of HIV testing and antiretroviral treatment in representative samples of FSWs and NSWs from the same population—we are unaware of previous studies which have done this.

► A limitation of our study is that our data were gathered between 2009 and 2011.

worldwide.[1–3] Though the United Nations Programme on HIV/AIDS (UNAIDS) has set ambitious '90-90-90' targets for the HIV care cascade (ie, HIV diagnosis, ART initiation and ART adherence—as a proxy for viral load suppression),[4] these are national-level targets, and it is necessary to consider how they can be implemented for key populations such as female sex workers (FSWs)—women who engage in commercial sex work or who exchange sex for goods or services.[5] HIV prevalence among FSWs in sub-Saharan Africa is estimated to be 10–20 times higher than in women in the general population.[6] Adequate access to HIV treatment for FSWs has the potential to improve the survival and health of FSWs, to reduce the risk of transmission to their partners, and to potentially alter population-level HIV incidence.[7] Therefore, reaching and exceeding UNAIDS targets among FSWs should be a primary objective for all national HIV control programmes.[8]

## INTRODUCTION

Achieving high antiretroviral treatment (ART) uptake for people living with HIV (PLHIV) is key to ending the HIV epidemic

Such large disparities in health between FSWs and non-sex worker (NSW) women support the need for specialist sex worker services, yet treatment coverage for FSWs remains poorly characterised.[9 10] It is also unclear whether inequalities for service access exist and at what stage of the HIV treatment cascade to focus more effort in driving uptake. While stigma, marginalisation and abuse of human rights have all been highlighted as significant barriers that can prevent FSWs from accessing HIV testing and treatment services,[11] relatively few studies exist on HIV treatment cascades among representative samples of FSWs. These include a study by Cowan and colleagues[12] in three urban sites in Zimbabwe (Victoria Falls, Hwange and Mutare) where 50%–70% were seropositive, of whom only 50% had been diagnosed. Of those diagnosed, 50%–70% had been initiated onto treatment, but due to the low rate of diagnosis, only 25%–35% of *all* seropositive FSWs in the study had received ART.[12] Still, very little is known about FSW in more rural settings or about how FSWs' use of HIV services compares with that of NSWs living in the same areas. A further unknown is the extent to which differences in heath service uptake between FSWs and NSWs reflect largely psychosocial factors resulting from involvement in sex work (eg, personal risk perception) as distinct from background socio-demographic factors associated with being involved in sex work in the first place (eg, marriage breakdown).

This study has the following aims: (1) to construct and compare HIV treatment cascades for FSWs and NSWs in a common, rural population; (2) to identify the background socio-demographic characteristics associated with involvement in commercial sex work in this population and (3) to identify the intermediate factors that might explain differences in health service uptake (testing and treatment) between FSWs and NSWs. To achieve these aims, we develop a new theoretical framework and test hypothesised determinants based on this framework using a unique data set which combines data from a general population household survey in four locations in Manicaland province, east Zimbabwe, with data from a parallel study of local FSWs conducted in the same locations using snowball sampling.

## METHODS
### Theoretical framework
Influenced by Boerma and Weir's proximate determinants model of HIV infection and mortality[13 14] and structural determinants frameworks of HIV among sex workers,[15] we developed a theoretical framework to explain the roles that involvement in sex work and its consequences can play in mediating associations between underlying socio-demographic characteristics and use of HIV testing and treatment services (figure 1 and online supplementary figure 1). It is hypothesised that, within any given sociocultural context, underlying socio-demographic characteristics contribute to whether or not a woman engages in sex work which may, in turn, alter her pattern of use of HIV services. In the framework, sex work is hypothesised to influence use of HIV services primarily through its effects on intermediate determinants that exist in four domains: personal, interpersonal, social and structural.

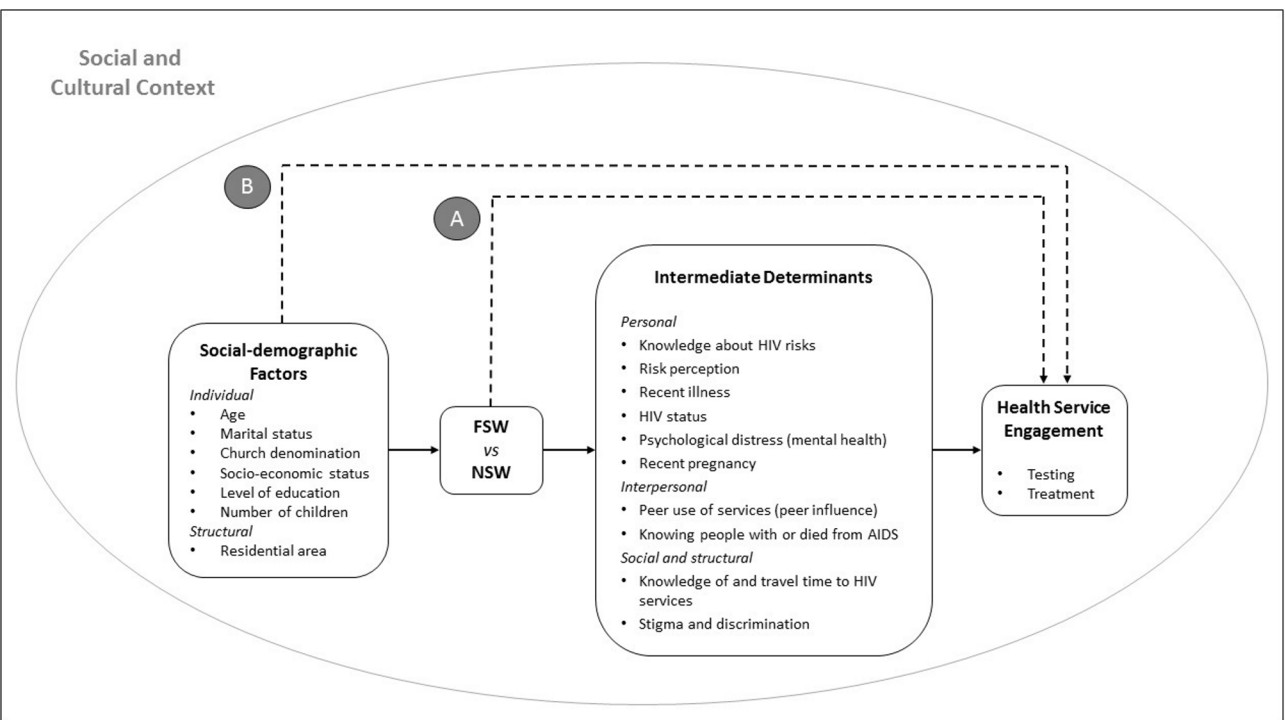

**Figure 1** Theoretical framework illustrating how engaging in sex work (or not) may influence use of HIV testing and treatment services. FSW, female sex workers; NSW, non-sex workers.

## Data

Data for this study were taken from the Manicaland HIV/STD Prevention Project (Manicaland study)[16] and the Manicaland Women at Risk Study (WR study).[17] The Manicaland study is an open-cohort general-population survey which examines the dynamics of HIV transmission and its impact in 12 sites in Manicaland province in eastern Zimbabwe (http://www.manicalandhivproject.org/). These sites represent four of the main socioeconomic strata in Manicaland: small towns, agricultural estates, roadside trading centres and subsistence farming villages. Topics covered in individual interviews included socioeconomic characteristics, sexual behaviour, psychosocial characteristics and use of HIV testing and treatment services. Participants were also requested to provide a dried blood sample for HIV sero-testing. The data used in this analysis were taken from the fifth round of the Manicaland survey (October 2009–July 2011) and were restricted to the four sites (one in each socioeconomic stratum) also covered by the WR study.

The data from the Manicaland study were linked with data from the WR study, a parallel targeted cohort study conducted to identify women at heightened risk of HIV infection through exchange of sex (including sex work), to enhance detection of FSWs and to permit comparison of HIV treatment cascades between FSWs and NSWs within a common wider population. The WR study is a research project, conducted in four of the same sites covered in the Manicaland study, which aimed to explore the sexual behaviours of women at heightened risk of HIV infection (http://www.manicalandhivproject.org/women-at-risk.html). Data for the WR study were collected between March 2010 and July 2011 using a combination of Priorities for Local AIDS Control Effort (PLACE, a form of location-based sampling)[18] and snowball sampling[19] methods. Data collection procedures have been described in detail elsewhere[20] but are summarised here. PLACE involves sampling locations of known sex work activity. An inventory of locations was created based on discussions with community members. Since only a small number of venues were identified, all venues were sampled. To capture exchange sex outside of specific venues, the population was sampled using a modified respondent-driven sampling approach.[21] Seeds were selected to represent the diversity of those involved in exchange sex. These seeds then recruited up to three peers that met broad eligibility criteria (women aged 18+ who had ever exchanged sex for money, goods or favours) and were compensated with one bar of laundry soap per respondent referred and invited to interview. We mitigated duplication and impersonation by cross-referencing names of nominated individuals with the names of women appearing to interview and by close monitoring by key informants (women with personal experience of sex work or who worked closely with women selling sex in the communities as health and support workers).

The FSWs who participated in both the Manicaland and WR study were requested to provide permission to link their data across both projects. Data for consenting participants were linked via probabilistic matching based on participant name, date of birth and village name.

## Study variables

### Female sex workers

The Manicaland and WR studies contained identical indicators of sex work. Informed by prior qualitative work within study communities[22] and in line with UNAIDS definitions,[23] participants in the Manicaland study were considered to be FSWs if they (1) self-identified as a sex worker or sex worker, (2) had ever gone to bars/beer halls to meet clients or (3) had exchanged sex for money/goods, in at least one of the two studies.

### Non-sex workers

NSWs in the study were taken to be all women interviewed in the Manicaland study who reported having ever had sex and who did not meet the definition of an FSW given above based on their self-reports in the Manicaland study and/or in the WR study.

FSWs and NSWs as defined above were included in the current analysis if they were aged 15–58 years (the age range covered by the WR study) and were resident in one of the four Manicaland study areas also covered by the WR study.

### HIV treatment cascade

HIV diagnosis was defined as the percentage of all HIV-positive participants (based on HIV tests done in the Manicaland study) who reported ever having been tested and having collected their results and received a positive result at their most recent HIV test. ART initiation was defined as the percentage of HIV-positive participants who knew their status (denominator) and also reported taking drugs 'that stop HIV from causing AIDS' (numerator). ART adherence was used as a further indicator of HIV service use and as a proxy for viral load suppression. HIV-positive participants who reported ever having started ART were included in the denominator; those who reported never having stopped or forgotten to take their medication and who reported taking ARVs regularly were included in the numerator.

### Health service uptake

Two measures of health service engagement were considered as dependent variables in our regression analyses: (1) uptake of HIV testing and (2) uptake of ART. Uptake of HIV testing was defined as ever having had an HIV test and collected the result. Uptake of treatment was measured in seropositive participants and based on reports of having taking drugs 'that stop HIV from causing AIDS'.

### Socio-demographic characteristics

Age,[24] marital status,(35) socioeconomic status (SES), religion,[24 25] area of residence,[24 26] education level,(15) and number of living children were considered as potential underlying determinants of involvement in sex work and use of HIV services (figure 1; see also more detailed explanation in online supplementary material). For SES,

we used a continuous combined measure of sellable and non-sellable assets,[27] divided into terciles (1=poorest → 3=richest). For religious denomination, we used Manzou's four-category grouping of Manicaland churches.[25]

### Intermediate determinants of HIV service uptake

Personal factors potentially mediating HIV service uptake in the theoretical framework included recent ill-health (self-reported experience of recent ill-health and whether or not this was believed to be HIV-related), self-reported symptoms of sexually transmitted diseases, self-reported recent pregnancies (that could translate to HIV testing through uptake of prevention of mother-to-child transmission (PMTCT) - services), HIV knowledge (number of correct responses to four questions: 0–2 correct answers=poor knowledge, 3–4 correct answers=good knowledge), HIV risk perception (whether participants perceived they had ever been at risk of becoming infected with HIV, and if so, was it through their own risky behaviour, their partner's risky behaviour or for other reasons), awareness of treatment for HIV and an objective mental health assessment using a locally validated questionnaire (Shona Symptom Questionnaire).[28 29] Interpersonal factors included HIV salience (number of people known by the participants who are living with HIV or who had died from AIDS) and awareness of other people using ART (individuals who were unaware of ART were combined with those unaware of anyone using ART because of small numbers). Potential social and structural influences included accessibility of ART or HIV Testing and Councelling (HTC) services; participant's awareness of a health facility offering HTC (or ART) and the estimated the travel time to the nearest health facility. Stigma was measured using two dichotomous variables: whether the participant was ever deterred from getting a test due to stigma or discrimination, and whether the participant felt that PLHIV faced stigma and discrimination within the community.

Travel time and stigma relating to HTC and awareness of ART were used only in the analysis of uptake of testing (ie, not for ART uptake). Travel time to ART services was used only in the analysis of ART uptake (ie, not for HIV testing).

### Statistical analyses

The analysis consisted of several stages. First, HIV prevalence and HIV treatment cascade outcomes were calculated and compared between FSWs and NSWs. Second, bivariate (age-adjusted) regression models were used to explore associations between socio-demographic characteristics and involvement in sex work, and associations between sex work and intermediate determinants of uptake hypothesised in the theoretical framework. Third, age-adjusted bivariate examination of associations between both socio-demographic characteristics and intermediate determinants and HIV testing and treatment was conducted to detect significant associations at P<0.1. Fourth, age-adjusted multivariable regression models were used to compare uptake of health services

in FSWs versus NSWs—before and after inclusion of socio-demographic factors and intermediate determinants of uptake (significant at P<0.1). All analyses were done using Stata V.14.

## RESULTS

### Identification of FSWs

In total, 174 participants were identified as FSWs in at least one of the two studies; 132 were included from the WR study (111 were identified based on their responses to the WR study questionnaire alone and 31 also self-identified as sex workers in the Manicaland questionnaire), and 32 were identified based on their answers to the Manicaland study questionnaire alone.

A total of 3402 women aged 15–59 years participated in the Manicaland study in round five in the four sites also covered by the WR study and were not identified as FSWs in either study. These participants were all treated as being NSWs; however, 135 were excluded from the study as they were outside the age range of the FSWs in the study (19–58 years), and a further 538 were excluded because they had not started sex. This produced a total sample of 2729 (FSWs=174, 6.4%; NSWs=2555, 93.6%).

### HIV prevalence

HIV prevalence was significantly higher in FSWs (52.6%, 95% CI 45.1% to 60.0%; n/n=91/173) compared with NSWs (19.8%, 18.3%–21.4%; 502/2535) (age-adjusted OR (AOR) 4.0; 2.9–5.5). Study HIV laboratory test results were inconclusive for 1 FSW and 2 NSWs.

### HIV treatment cascades in FSWs and NSWs

In HIV-positive women, diagnosis (ie, women who were aware of their HIV-positive status) was higher in FSWs (58.2%, 95% CI 47.7% to 68.1%; 53/91) than in NSWs (42.6%, 38.3%–47.9%; 214/502) (AOR 1.62; 1.02–2.59; P=0.042). In HIV-negative women, there was no significant difference between FSWs and NSWs in uptake of HIV testing (81.7% in FSWs; 75.3% in NSWs; AOR 1.40; 0.78–2.51; P=0.259).

In women diagnosed with HIV infection, initiation onto ART was higher in FSWs (84.9%, 72.1%–92.4%; 45/53) than in NSWs (64.2%, 57.5%–70.4%; 138/214) (AOR 2.33; 1.03–5.28; P=0.043). No significant difference was found between FSWs (91.1%, 77.9%–96.7%; 41/45) and NSWs (90.5%, 84.2%–94.4%; 124/137) in self-reported adherence to ART (AOR 1.08; 0.33–3.52; P=0.901). Overall, 49.4% (45/91) of *all* HIV-positive FSWs reported being on and adhering to ART compared with 27.5% (139/505) of infected NSWs (as shown in figure 2A, B).

### Socio-demographic characteristics and intermediate determinants of HIV service use associated with sex work

Table 1 shows the age-adjusted bivariate associations between socio-demographic factors and sex work, and between sex work and intermediate determinants; that is, the first two pathways in the theoretical framework

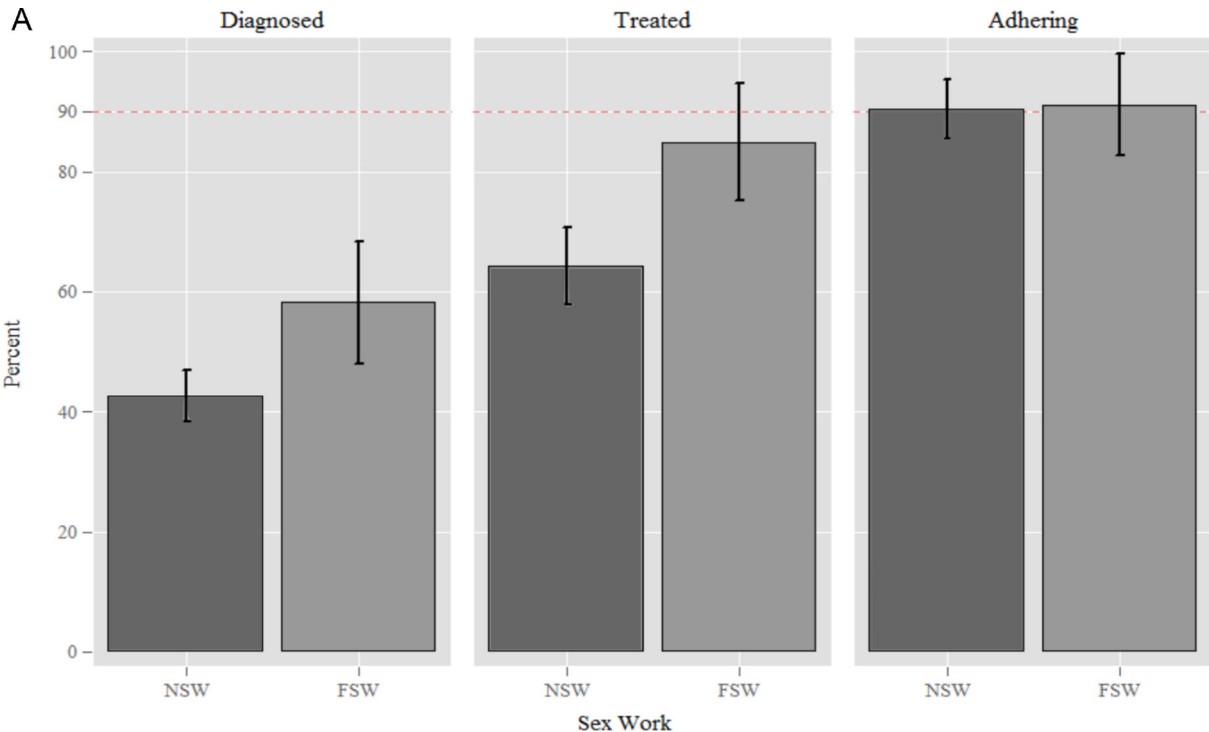

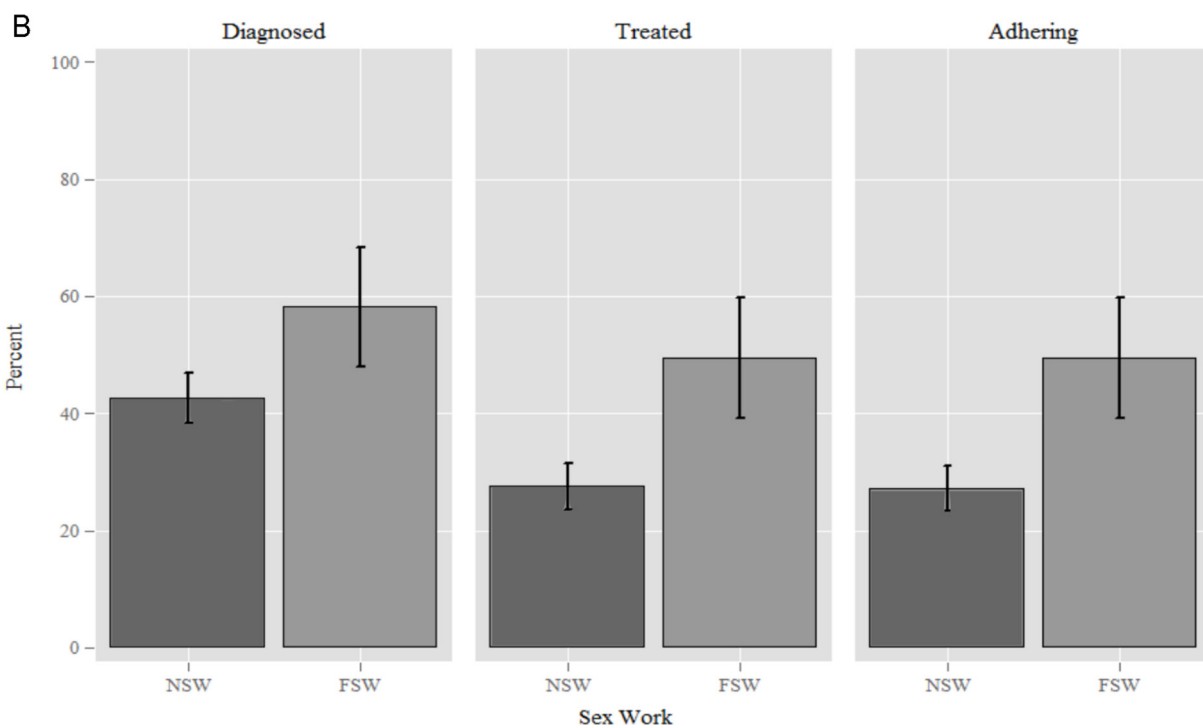

**Figure 2** (A, B) Comparison of HIV treatment cascades (non-cumulative and cumulative) for female sex workers (FSWs) and non-sex workers (NSWs) in Manicaland, Zimbabwe, 2009–2011. Figures illustrating the proportion of FSWs and NSWs who achieve optimal outcomes at each stage of the cascade. (A) The proportions of HIV-positive women who have been diagnosed, the proportions of treated among those who have been diagnosed and the proportions adhering to their medication among those who have been treated. A 90% reference line is included to illustrate UNAIDS targets. (B) The denominator is all HIV-positive women at each stage of the cascade.

(figure 1). Sex work was most common in women aged 30–49 years; single, divorced and widowed women; women with no religious affiliation; women in the two poorest terciles; women living in small towns and women with no living children. For the intermediate determinants, sex work was associated with greater risk perception for HIV infection (particularly through personal risky behaviours); knowing at least three people with HIV;

**Table 1** Socio-demographic characteristics associated with female involvement in sex work, and associations between sex work and intermediate determinants of HIV testing and treatment, Manicaland Zimbabwe, 2009–2011

| | FSWs | | NSWs | | | | |
| --- | --- | --- | --- | --- | --- | --- | --- |
| | n | % | n | % | N | AOR | 95% CI |
| **Socio-demographic characteristic** | | | | | | | |
| Age-group (years) | | | | | | | |
| 19–29 | 36 | 20.7 | 950 | 37.2 | 986 | 1 | – |
| 30–39 | 74 | 42.5 | 710 | 27.8 | 784 | 2.75 | 1.83 to 4.14 |
| 40–49 | 46 | 26.4 | 527 | (20.6) | 573 | 2.3 | 1.47 to 3.61 |
| 50–58 | 18 | 10.3 | 368 | 14.4 | 386 | 1.29 | 0.72 to 2.30 |
| Marital status | | | | | | | |
| Never married | 8 | 4.6 | 68 | 2.7 | 76 | 3.18 | 1.46 to 6.92 |
| Married | 87 | 50.0 | 1912 | 74.8 | 1999 | 1 | – |
| Divorced or separated | 41 | 23.6 | 225 | 8.8 | 266 | 3.76 | 2.52 to 5.62 |
| Widowed | 37 | 21.3 | 350 | 13.7 | 387 | 2.25 | 1.47 to 3.45 |
| Church denomination | | | | | | | |
| Christian | 89 | 51.1 | 1385 | 54.2 | 1474 | 1 | – |
| Spiritual | 53 | 30.5 | 882 | 34.5 | 935 | 0.91 | 0.64 to 1.30 |
| Other | 22 | 12.6 | 253 | 9.9 | 275 | 1.35 | 0.83 to 2.21 |
| None | 10 | 5.7 | 33 | 1.3 | 43 | 4.47 | 2.10 to 9.52 |
| Socioeconomic status | | | | | | | |
| First (poorest) tercile | 121 | 69.5 | 1635 | 64.0 | 1756 | 1 | – |
| Second tercile | 42 | 24.1 | 572 | 22.4 | 614 | 0.97 | 0.67 to 1.40 |
| Third tercile | 7 | 4.0 | 263 | 10.3 | 270 | 0.37 | 0.17 to 0.80 |
| Residential area | | | | | | | |
| Town | 64 | 36.8 | 597 | 23.4 | 661 | 1 | – |
| Agricultural estate | 40 | 23.0 | 610 | 23.9 | 650 | 0.59 | 0.39 to 0.89 |
| Roadside settlement | 45 | 25.9 | 702 | 27.5 | 747 | 0.59 | 0.40 to 0.89 |
| Subsistence farming village | 25 | 14.4 | 646 | 25.3 | 671 | 0.36 | 0.22 to 0.58 |
| Education | | | | | | | |
| Primary or none | 74 | 42.5 | 944 | 36.9 | 1018 | 1 | – |
| Secondary or higher | 100 | 57.5 | 1611 | 63.1 | 1711 | 0.78 | 0.54 to 1.11 |
| Children alive | | | | | | | |
| None | 44 | 25.3 | 365 | 14.3 | 409 | 1 | – |
| 1 | 45 | 25.9 | 750 | 29.4 | 795 | 0.54 | 0.34 to 0.83 |
| 2 | 43 | 24.7 | 701 | 27.4 | 744 | 0.46 | 0.29 to 0.72 |
| 3 | 20 | 11.5 | 436 | 17.1 | 456 | 0.27 | 0.15 to 0.47 |
| 4 | 22 | 12.6 | 303 | 11.9 | 325 | 0.39 | 0.22 to 0.67 |
| **Intermediate determinants** | | | | | | | |
| HIV testing | | | | | | | |
| HIV result | | | | | | | |
| Positive | 91 | 52.3 | 505 | 19.8 | 596 | 4.00 | 2.90 to 5.50 |
| Negative | 82 | 47.1 | 2048 | 80.2 | 2130 | 1 | – |
| Knowledge about HIV risks | | | | | | | |
| Good | 158 | 90.8 | 2167 | 84.8 | 2325 | 1.63 | 0.96 to 2.76 |
| Poor | 16 | 9.2 | 388 | 15.2 | 404 | 1 | – |

Continued

**Table 1** Continued

| | FSWs | | NSWs | | | | |
|---|---|---|---|---|---|---|---|
| | n | % | n | % | N | AOR | 95% CI |
| **Knowing persons living with or who PLHIV/died from HIV** | | | | | | | |
| 0 | 14 | 8.0 | 506 | 19.8 | 520 | 1 | – |
| 1–2 | 22 | 12.6 | 455 | 17.8 | 477 | 1.69 | 0.85 to 3.36 |
| 3–4 | 29 | 16.7 | 449 | 17.6 | 478 | 2.13 | 1.11 to 4.09 |
| 5–6 | 33 | 19.0 | 458 | 17.9 | 491 | 2.43 | 1.28 to 4.61 |
| 7 | 76 | 43.7 | 687 | 26.9 | 763 | 3.56 | 1.98 to 6.38 |
| **Risk perception for HIV infection** | | | | | | | |
| Own high-risk behaviour | 39 | 22.4 | 41 | 1.6 | 80 | 18.82 | 11.49 to 30.81 |
| Partner(s)' high-risk behaviour | 18 | 10.3 | 166 | 6.5 | 184 | 2.18 | 1.28 to 3.73 |
| Other reasons | 21 | 12.1 | 138 | 5.4 | 159 | 3.39 | 2.05 to 5.63 |
| None | 96 | 55.2 | 2210 | 86.5 | 2306 | 1.00 | – |
| **STD symptoms in last 12 months** | | | | | | | |
| Yes | 29 | 16.7 | 215 | 8.4 | 244 | 2.05 | 1.34 to 3.13 |
| No | 145 | 83.3 | 2340 | 91.6 | 2485 | 1 | – |
| **Sickness in last 12 months** | | | | | | | |
| HIV-related illness | 23 | 13.2 | 92 | 3.6 | 115 | 4.09 | 2.41 to 6.93 |
| Other illness | 81 | 46.6 | 1125 | 44.0 | 1206 | 1.37 | 0.98 to 1.91 |
| None | 69 | 39.7 | 1335 | 52.3 | 1404 | 1 | – |
| **Psychological distress** | | | | | | | |
| Yes | 43 | 24.7 | 298 | 11.7 | 341 | 1.31 | 1.60 to 3.34 |
| No | 131 | 75.3 | 2257 | 88.3 | 2388 | 1 | – |
| **Pregnancies in last 3 years** | | | | | | | |
| One or more | 49 | 28.2 | 1077 | 42.2 | 1126 | 0.59 | 0.40 to 0.87 |
| None | 125 | 71.8 | 1478 | 57.8 | 1603 | 2 | – |
| **Stigma and discrimination (affecting testing)** | | | | | | | |
| Yes | 2 | 1.1 | 26 | 1.0 | 28 | 0.05 | 0.25 to 4.49 |
| No | 172 | 98.9 | 2529 | 99.0 | 2701 | 1 | – |
| **Travel time to HIV testing facility** | | | | | | | |
| <30 min | 61 | 35.1 | 390 | 15.3 | 451 | 1 | – |
| 30–59 min | 39 | 22.4 | 585 | 22.9 | 624 | 0.42 | 0.27 to 0.64 |
| 60–89 min | 23 | 13.2 | 587 | 23.0 | 610 | 0.24 | 0.15 to 0.40 |
| 90 min | 48 | 27.6 | 849 | 33.2 | 897 | 0.35 | 0.23 to 0.52 |
| Uncertain | 3 | 1.7 | 144 | 5.6 | 147 | – | – |
| ART | | | | | | | |
| **Knowledge of ART** | | | | | | | |
| Yes | 126 | 72.4 | 1341 | 52.5 | 1467 | 1.35 | 1.67 to 3.31 |
| No | 48 | 27.6 | 1203 | 47.1 | 1251 | 1 | – |
| **Stigma and discrimination (in the community)** | | | | | | | |
| Yes | 43 | 24.7 | 462 | 18.1 | 505 | 0.46 | 1.01 to 2.08 |
| No | 131 | 75.3 | 2090 | 81.8 | 2221 | 1 | – |
| **Peer influence** | | | | | | | |
| Relative(s) on ART | 41 | 23.6 | 469 | 18.4 | 510 | 2.14 | 1.41 to 3.24 |
| Friend(s) on ART | 55 | 31.6 | 333 | 13.0 | 388 | 3.98 | 2.69 to 5.89 |
| None | 58 | 33.3 | 1496 | 58.6 | 1554 | 1 | – |

**Table 1** Continued

| | FSWs | | NSWs | | | | |
|---|---|---|---|---|---|---|---|
| | n | % | n | % | N | AOR | 95% CI |
| Travel time to ART service * | | | | | | | |
| <30 min | 34 | 19.5 | 172 | 6.7 | 206 | 1 | – |
| 30–59 min | 17 | 9.8 | 261 | 10.2 | 278 | 0.31 | 0.17 to 0.57 |
| 60–89 min | 17 | 9.8 | 224 | 8.8 | 241 | 0.38 | 0.20 to 0.70 |
| 90 min | 33 | 19.0 | 341 | 13.3 | 374 | 0.46 | 0.28 to 0.78 |
| Uncertain | 73 | 42.0 | 1557 | 60.9 | 1630 | – | – |

*Includes women not aware of HIV testing and ART services to prevent exclusion of these participants from the multivariable analysis. ORs were not interpreted for this group as they are not comparable with the reference category.

AOR, adjusted OR; ART, antiretroviral treatment; FSWs, female sex workers; NSWs, non-sex workers; STD, sexually transmitted diseases.

experiencing recent HIV-related illness and STD symptoms; poor mental health; no pregnancies in the past three years; short travel times to HTC and ART facilities; having heard of ART and reporting that HIV stigma and discrimination exist in the community. We also found a non-significant difference between FSWs and NSWs for knowledge of HIV risks.

### Socio-demographic characteristics and intermediate determinants of HIV testing

In HIV-positive and HIV-negative women combined, FSWs were more likely than NSWs to have ever been tested for HIV (FSWs: 81.6%; NSWs: 75.3%; AOR 1.50; 95% CI 1.00 to 2.24). Table 2 shows the bivariate and multivariable associations of socio-demographic factors and intermediate determinants on testing. In bivariate analysis, sex work was associated with ever having been tested, as are all socio-demographic factors included in this analysis. All intermediate determinants with the exception of psychological distress and stigma (after testing and in the community) are associated with testing at P<0.1. In multivariable analysis, the association between sex work and testing is strengthened after adjusting for socio-demographic factors, with sex work being associated with 75% increased odds of testing (AOR 1.75, 1.14–1.69). However, this association between sex work and testing disappears after also accounting for intermediate determinants (AOR 1.11, 0.69–1.81).

When the analysis was restricted to HIV-positive women (online supplementary table S1), the association between FSWs and diagnosis approached statistical significance after adjusting for socio-demographic factors (AOR 1.83, 1.00–3.37; P=0.052).

### Socio-demographic characteristics and intermediate determinants of HIV treatment

Table 3 shows the bivariate and multivariable associations between socio-demographic factors and intermediate determinants and ART initiation. In bivariate analysis, sex work is associated with 164% increased odds of ART initiation (AOR 2.64, 1.16–6.00). Older age, more urban site types, having no living children, knowing people who have/had HIV, not having psychological distress and shorter travel times to ART were also associated with treatment initiation at P<0.1. In multivariable analysis, FSWs still tended to have higher odds of ART initiation than NSWs, but this association was no longer statistically significant when socio-demographic factors were accounted for (AOR 2.28, 0.97–5.39).

### DISCUSSION

FSWs had higher uptake of HIV testing and ART services than other sexually experienced women in our study areas in east Zimbabwe. For HIV testing, this advantage strengthened after accounting for differences in background socio-demographic characteristics but disappeared after further adjustment for intermediate determinants, confirming a process of mediation hypothesised in the theoretical framework. FSWs' greater knowledge about HIV and greater personal perceived risk of being HIV-positive, their better knowledge of and proximity to testing services, and their greater likelihood of perceiving HIV-related symptoms (ie, the intermediate determinants outlined in our framework) may have contributed to their higher levels of HIV testing. Greater ART uptake in FSWs compared with NSWs was explained not by intermediate factors relating to sex work status but by their older ages (ie, fewer aged 19–29 years) and lower numbers of living children. The reason for the link with small numbers of living children is not clear, but, in *Shona* culture,[30] subfertility/infertility can lead to divorce which, in turn, is associated with greater likelihood of involvement in sex work. Widowhood at young ages may be associated with early HIV infection, reduced fertility, high early child mortality and involvement in sex work. Also, low fertility and early child mortality can be markers for more advanced HIV infection,[31] thereby increasing the likelihood of meeting the eligibility criteria for ART that pertained at the time of the study (CD4 <350 or WHO phases III or IV in 2009–2011). ART adherence was similar in FSWs and NSWs.

HIV prevalence in FSWs in east Zimbabwe (52.6%) was comparable with prevalence in FSWs in other southern African countries (range 59.6%–70.7%).[6] The proportion

**Table 2** Factors contributing to differences in uptake of HIV testing ever in lifetime between female sex workers (FSWs) and non-sex workers (NSWs), Manicaland, 2009–2011

| | Bivariate analysis | | | | Socio-demographic | Socio-demographic + sex work | | Intermediate Determinants | | Intermediate sex work | | Intermediate determinants + sex work | | Full model | |
|---|---|---|---|---|---|---|---|---|---|---|---|---|---|---|---|
| | n | % | AOR | 95% CI | AOR | AOR | 95% CI | AOR | 95% CI | AOR | 95% CI | AOR | 95% CI | AOR | 95% CI |
| **Female sex work** | | | | | | | | | | | | | | | |
| Sex work | | | | | | | | | | | | | | | |
| NSWs | 1925 | 93.1 | 1 | – | 1 | 1 | – | – | – | – | – | 1 | – | 1 | – |
| FSWs | 142 | 6.9 | 1.5 | 1.00 to 2.24 | – | 1.75 | 1.14 to 2.69 | – | – | – | – | 0.99 | 0.63 to 1.57 | 1.11 | 0.69 to 1.81 |
| **Socio-demographic** | | | | | | | | | | | | | | | |
| Age group (years) | | | | | | | | | | | | | | | |
| 19–29 | 822 | 39.8 | 1 | – | 1 | 1 | – | – | – | – | – | – | – | 1 | – |
| 30–39 | 622 | 30.1 | 0.75 | 0.59 to 0.95 | 0.69 | 0.67 | 0.53 to 0.90 | – | – | – | – | – | – | 0.65 | 0.48 to 0.89 |
| 40–49 | 402 | 19.4 | 0.46 | 0.36 to 0.59 | 0.46 | 0.45 | 0.34 to 0.61 | – | – | – | – | – | – | 0.54 | 0.37 to 0.78 |
| 50–58 | 221 | 10.7 | 0.27 | 0.20 to 0.35 | 0.29 | 0.29 | 0.21 to 0.41 | – | – | – | – | – | – | 0.39 | 0.26 to 0.59 |
| Marital status | | | | | | | | | | | | | | | |
| Never married | 283 | 13.7 | 0.28 | 0.17 to 0.45 | 0.31 | 0.3 | 0.18 to 0.50 | – | – | – | – | – | – | 0.43 | 0.24 to 0.78 |
| Married | 42 | 2.0 | 1 | – | 1 | 1 | – | – | – | – | – | – | – | 1 | – |
| Divorced or separated | 1551 | 75.0 | 0.78 | 0.58 to 1.05 | 0.8 | 0.77 | 0.56 to 1.05 | – | – | – | – | – | – | 0.83 | 0.58 to 1.17 |
| Widowed | 191 | 9.2 | 1.3 | 0.99 to 1.70 | 1.4 | 1.38 | 1.06 to 1.85 | – | – | – | – | – | – | 1.26 | 0.92 to 1.72 |
| Church denomination | | | | | | | | | | | | | | | |
| Christian | 1131 | 54.7 | 1 | – | 1 | 1 | – | – | – | – | – | – | – | 1 | – |
| Spiritual | 701 | 33.9 | 0.84 | 0.69 to 1.03 | 0.89 | 0.89 | 0.72 to 1.09 | – | – | – | – | – | – | 0.95 | 0.75 to 1.20 |
| Other | 203 | 9.8 | 0.78 | 0.58 to 1.06 | 0.87 | 0.87 | 0.63 to 1.20 | – | – | – | – | – | – | 0.94 | 0.65 to 1.34 |
| None | 30 | 1.5 | 0.56 | 0.28 to 1.09 | 0.56 | 0.52 | 0.28 to 1.13 | – | – | – | – | – | – | 0.43 | 0.20 to 0.93 |
| Socioeconomic status | | | | | | | | | | | | | | | |
| First (poorest) tercile | 1308 | 63.3 | 1 | – | 1 | 1 | – | – | – | – | – | – | – | 1 | – |
| Second tercile | 472 | 22.8 | 1.13 | 0.90 to 1.41 | 1.04 | 1.04 | 0.83 to 1.30 | – | – | – | – | – | – | 0.94 | 0.73 to 1.22 |
| Third tercile | 219 | 10.6 | 1.35 | 0.97 to 1.87 | 1.06 | 1.1 | 0.74 to 1.53 | – | – | – | – | – | – | 0.9 | 0.60 to 1.34 |
| Residential area | | | | | | | | | | | | | | | |
| Town | 542 | 26.2 | 1 | – | 1 | 1 | – | – | – | – | – | – | – | 1 | – |
| Agricultural estate | 460 | 22.3 | 0.57 | 0.44 to 0.74 | 0.59 | 0.61 | 0.44 to 0.79 | – | – | – | – | – | – | 0.63 | 0.45 to 0.88 |
| Roadside settlement | 569 | 27.5 | 0.81 | 0.62 to 1.06 | 0.79 | 0.81 | 0.59 to 1.05 | – | – | – | – | – | – | 0.94 | 0.68 to 1.30 |

Continued

**Table 2**  Continued

| | Bivariate analysis | | | | Socio-demographic | | Socio-demographic + sex work | | Intermediate Determinants | | Intermediate determinants + sex work | | Full model | |
|---|---|---|---|---|---|---|---|---|---|---|---|---|---|---|
| | n | % | AOR | 95% CI | AOR | 95% CI | AOR | 95% CI | AOR | 95% CI | AOR | 95% CI | AOR | 95% CI |
| Subsistence farming village | 496 | 24.0 | 0.71 | 0.54 to 0.92 | 0.7 | 0.52 to 0.95 | 0.73 | 0.55 to 0.99 | – | – | – | – | 0.93 | 0.67 to 1.31 |
| **Education** | | | | | | | | | | | | | | |
| Primary or less | 675 | 32.7 | 1 | – | 1 | – | 1 | – | – | – | – | – | 1 | – |
| Secondary or higher | 1392 | 67.3 | 1.47 | 1.18 to 1.83 | 1.45 | 1.15 to 1.82 | 1.46 | 1.16 to 1.83 | – | – | – | – | 1.06 | 0.81 to 1.37 |
| **Children alive** | | | | | | | | | | | | | | |
| None | 257 | 12.4 | 1 | – | 1 | – | 1 | – | – | – | – | – | 1 | – |
| 1 | 632 | 30.6 | 1.9 | 1.45 to 2.50 | 1.63 | 1.23 to 2.16 | 1.67 | 1.26 to 2.22 | – | – | – | – | 1.52 | 1.10 to 2.09 |
| 2 | 586 | 28.4 | 1.79 | 1.35 to 2.36 | 1.53 | 1.14 to 2.05 | 1.57 | 1.17 to 2.11 | – | – | – | – | 1.48 | 1.06 to 2.06 |
| 3 | 343 | 16.6 | 1.6 | 1.18 to 2.18 | 1.43 | 1.04 to 1.98 | 1.5 | 1.08 to 2.07 | – | – | – | – | 1.45 | 1.01 to 2.08 |
| 4 | 249 | 12.0 | 1.88 | 1.33 to 2.65 | 1.73 | 1.21 to 2.48 | 1.79 | 1.24 to 2.57 | – | – | – | – | 1.72 | 1.14 to 2.59 |
| **Intermediate determinants** | | | | | | | | | | | | | | |
| **HIV result** | | | | | | | | | | | | | | |
| Positive | 454 | 22.0 | 1.09 | 0.88 to 1.36 | – | – | – | – | – | – | – | – | – | – |
| Negative | 1611 | 77.9 | 1 | – | – | – | – | – | – | – | – | – | – | – |
| **Knowledge about HIV risks** | | | | | | | | | | | | | | |
| Good | 1787 | 86.5 | 1.5 | 1.18 to 1.90 | – | – | – | – | 1.42 | 1.08 to 1.86 | 1.42 | 1.08 to 1.86 | 1.35 | 1.02 to 1.80 |
| Poor | 280 | 13.5 | 1 | – | – | – | – | – | 1 | – | 1 | – | 1 | – |
| **Risk perception for HIV infection** | | | | | | | | | | | | | | |
| Own high-risk behaviour | 68 | 3.3 | 2.36 | 1.26 to 4.43 | – | – | – | – | 1.21 | 0.62 to 2.36 | 1.21 | 0.60 to 2.43 | 1.3 | 0.63 to 2.70 |
| Partner(s)' high-risk behaviour | 172 | 8.3 | 6.21 | 3.41 to 11.29 | – | – | – | – | 4.53 | 2.30 to 8.94 | 4.53 | 2.30 to 8.94 | 4.61 | 2.29 to 9.29 |
| Other reasons | 133 | 6.4 | 2.12 | 1.37 to 3.29 | – | – | – | – | 1.47 | 0.92 to 2.35 | 1.47 | 0.92 to 2.35 | 1.41 | 0.87 to 2.29 |
| None | 1694 | 82.0 | 1 | – | – | – | – | – | 1 | – | 1 | – | 1 | – |
| **Knowing persons living with or who PLHIV/died from HIV** | | | | | | | | | | | | | | |
| 0 | 344 | 16.6 | 1 | – | – | – | – | – | 1 | – | 1 | – | 1 | – |
| 1–2 | 350 | 16.9 | 1.51 | 1.14 to 2.00 | – | – | – | – | 1.14 | 0.83 to 1.57 | 1.14 | 0.83 to 1.57 | 1.12 | 0.80 to 1.57 |
| 3–4 | 392 | 19.0 | 2.56 | 1.89 to 3.47 | – | – | – | – | 2.09 | 1.48 to 2.95 | 2.09 | 1.48 to 2.95 | 2.04 | 1.43 to 2.91 |
| 5–6 | 389 | 18.8 | 2.26 | 1.69 to 3.04 | – | – | – | – | 1.53 | 1.11 to 2.12 | 1.53 | 1.11 to 2.12 | 1.47 | 1.05 to 2.06 |
| 7 | 592 | 28.6 | 2.07 | 1.60 to 2.69 | – | – | – | – | 1.35 | 1.01 to 1.82 | 1.35 | 1.01 to 1.82 | 1.27 | 0.93 to 1.72 |

Continued

**Table 2** Continued

| | Bivariate analysis | | | | Socio-demographic | | Socio-demographic + sex work | | Intermediate Determinants | | Intermediate determinants + sex work | | Full model | |
| --- | --- | --- | --- | --- | --- | --- | --- | --- | --- | --- | --- | --- | --- | --- |
| | n | % | AOR | 95% CI | AOR | 95% CI | AOR | 95% CI | AOR | 95% CI | AOR | 95% CI | AOR | 95% CI |
| STD symptoms in last 12 months | | | | | | | | | | | | | | |
| Yes | 195 | 9.4 | 1.5 | 1.00 to 2.24 | – | – | – | – | 0.83 | 0.57 to 1.20 | 0.83 | 0.57 to 1.20 | 0.83 | 0.56 to 1.22 |
| No | 1872 | 90.6 | 1 | – | – | – | – | – | 1 | – | 1 | – | 1 | – |
| Sickness in last 12 months | | | | | | | | | | | | | | |
| HIV-related illness | 109 | 5.3 | 8.08 | 3.50 to 18.67 | – | – | – | – | 3.78 | 1.55 to 9.18 | 3.78 | 1.55 to 9.18 | 4.35 | 1.76 to 10.71 |
| Other illness | 904 | 43.7 | 1.04 | 0.87 to 1.25 | – | – | – | – | 0.97 | 0.80 to 1.19 | 0.97 | 0.80 to 1.19 | 0.97 | 0.78 to 1.20 |
| None | 1052 | 50.9 | 1 | – | – | – | – | – | 1 | – | 1 | – | 1 | – |
| Psychological distress | | | | | | | | | | | | | | |
| Yes | 259 | 12.5 | 1.12 | 0.85 to 1.47 | – | – | – | – | – | – | – | – | – | – |
| No | 1808 | 87.5 | 1 | – | – | – | – | – | – | – | – | – | – | – |
| Pregnancies in last 3 years | | | | | | | | | | | | | | |
| One or more | 974 | 47.1 | 2.32 | 1.82 to 2.96 | – | – | – | – | 2.51 | 1.91 to 3.28 | 2.5 | 1.91 to 3.28 | 2.42 | 1.82 to 3.22 |
| None | 1093 | 52.9 | 1 | – | – | – | – | – | 1 | – | 1.00 | – | 1 | – |
| Travel time to HIV testing facility * | | | | | | | | | | | | | | |
| <30 min | 385 | 18.6 | 1 | – | – | – | – | – | 1 | – | 1 | – | 1 | – |
| 30–59 min | 508 | 24.6 | 0.78 | 0.56 to 1.09 | – | – | – | – | 0.73 | 0.51 to 1.03 | 0.73 | 0.51 to 1.03 | 0.73 | 0.51 to 1.06 |
| 60–89 min | 470 | 22.7 | 0.63 | 0.45 to 0.87 | – | – | – | – | 0.58 | 0.41 to 0.82 | 0.58 | 0.41 to 0.82 | 0.5 | 0.35 to 0.73 |
| 90 min | 690 | 33.4 | 0.63 | 0.46 to 0.86 | – | – | – | – | 0.58 | 0.42 to 0.80 | 0.58 | 0.42 to 0.80 | 0.47 | 0.33 to 0.67 |
| Uncertain | 14 | 0.7 | – | – | – | – | – | – | – | – | – | – | – | – |
| Knowledge of ART | | | | | | | | | | | | | | |
| Yes | 1224 | 59.2 | 2.38 | 1.98 to 2.86 | – | – | – | – | 1.51 | 1.23 to 1.87 | 1.51 | 1.23 to 1.87 | 1.48 | 1.19 to 1.85 |
| No | 833 | 40.3 | 1 | – | – | – | – | – | 1 | – | 1 | – | 1 | – |
| Stigma and discrimination (affecting testing) | | | | | | | | | | | | | | |
| Yes | 19 | 0.9 | 0.76 | 0.34 to 1.73 | – | – | – | – | – | – | – | – | – | – |
| No | 2048 | 99.1 | 1 | – | – | – | – | – | – | – | – | – | – | – |
| Stigma and discrimination (in the community) | | | | | | | | | | | | | | |
| Yes | 395 | 19.1 | 1.18 | 0.93 to 1.50 | – | – | – | – | – | – | – | – | – | – |
| No | 1669 | 80.7 | 1 | – | – | – | – | – | – | – | – | – | – | – |

*Includes women not aware of HIV testing and ART services to prevent exclusion of these participants from the multivariable analysis. ORs were not interpreted for this group as they are not comparable with the reference category.

AOR, adjusted OR; ART, antiretroviral treatment; STD, sexually transmitted diseases.

**Table 3** Factors contributing to differences in uptake of antiretroviral treatment between female sex workers (FSWs) and non-sex workers (NSWs), Manicaland, Zimbabwe, 2009–2011

| | Bivariate analysis | | | | Socio-demographic | | Socio-demographic sex work | | Intermediate determinants | | Intermediate determinants sex work | | Full model | |
|---|---|---|---|---|---|---|---|---|---|---|---|---|---|---|
| | n | % | AOR | 95% CI | AOR | 95% CI | AOR | 95% CI | AOR | 95% CI | AOR | 95% CI | AOR | 95% CI |
| **Female sex work** | | | | | | | | | | | | | | |
| Sex work | | | | | | | | | | | | | | |
| NSWs | 138 | 75.4 | 1 | – | – | – | 1 | – | – | – | 1 | – | 1 | – |
| FSWs | 45 | 24.6 | 2.64 | 1.16 to 6.00 | – | – | 2.28 | 0.97 to 5.39 | – | – | 3.46 | 0.91 to 13.16 | 3.51 | 0.79 to 15.47 |
| **Socio-demographic** | | | | | | | | | | | | | | |
| Age | | | | | | | | | | | | | | |
| Age (continuous) | – | – | 1.53 | 1.14 to 2.06 | 1.62 | 1.19 to 2.22 | 1.57 | 1.14 to 2.15 | – | – | – | – | 1.63 | 1.03 to 2.57 |
| Age² | – | – | 1 | 0.99 to 1.00 | 0.99 | 0.99 to 1.00 | 1 | 0.99 to 1.00 | – | – | – | – | 1 | 0.99 to 1.00 |
| Marital status | | | | | | | | | | | | | | |
| Never married | 2 | 1.1 | 0.33 | 0.04 to 2.57 | – | – | – | – | – | – | – | – | – | – |
| Married | 72 | 39.3 | 1 | – | – | – | – | – | – | – | – | – | – | – |
| Divorced/separated | 29 | 15.8 | 1.56 | 0.66 to 3.65 | – | – | – | – | – | – | – | – | – | – |
| Widowed | 80 | 43.7 | 1.41 | 0.73 to 2.74 | – | – | – | – | – | – | – | – | – | – |
| Church denomination | | | | | | | | | | | | | | |
| Christian churches | 100 | 54.6 | 1 | – | – | – | – | – | – | – | – | – | – | – |
| Spiritual churches | 59 | 32.2 | 0.62 | 0.35 to 1.12 | – | – | – | – | – | – | – | – | – | – |
| Other religion | 19 | 10.4 | 1.29 | 0.46 to 3.58 | – | – | – | – | – | – | – | – | – | – |
| No religion | 5 | 2.7 | 2 | 0.30 to 13.48 | – | – | – | – | – | – | – | – | – | – |
| Socioeconomic status | | | | | | | | | | | | | | |
| 1 (poorest) | 113 | 61.7 | 1 | – | 1 | – | 1 | – | – | – | – | – | – | – |
| 2 | 47 | 25.7 | 1.21 | 0.63 to 2.34 | – | – | – | – | – | – | – | – | – | – |
| 3 | 20 | 10.9 | 2.3 | 0.82 to 6.44 | – | – | – | – | – | – | – | – | – | – |
| Residential area | | | | | | | | | | | | | | |
| Town | 62 | 33.9 | 1 | – | 1 | – | 1 | – | – | – | – | – | 1 | – |
| Agricultural estate | 54 | 29.5 | 1.1 | 0.52 to 2.35 | 1.27 | 0.58 to 2.76 | 1.37 | 0.62 to 3.01 | – | – | – | – | 2.53 | 0.77 to 8.29 |
| Roadside settlement | 35 | 19.1 | 0.48 | 0.23 to 1.01 | 0.54 | 0.26 to 1.15 | 0.6 | 0.28 to 1.28 | – | – | – | – | 0.87 | 0.29 to 2.61 |
| Subsistence farming village | 32 | 17.5 | 0.82 | 0.37 to 1.82 | 0.96 | 0.42 to 2.16 | 1.01 | 0.44 to 2.30 | – | – | – | – | 1.78 | 0.53 to 5.95 |

Continued

**Table 3** Continued

| | Bivariate analysis | | | | Socio-demographic | | Socio-demographic sex work | | Intermediate determinants | | Intermediate determinants sex work | | Full model | |
|---|---|---|---|---|---|---|---|---|---|---|---|---|---|---|
| | n | % | AOR | 95% CI | AOR | 95% CI | AOR | 95% CI | AOR | 95% CI | AOR | 95% CI | AOR | 95% CI |
| **Education** | | | | | | | | | | | | | | |
| Primary or less | 86 | 47.0 | 1 | – | – | – | – | – | – | – | – | – | – | – |
| Secondary or higher | 97 | 53.0 | 1.09 | 0.58 to 2.07 | – | – | – | – | – | – | – | – | – | – |
| **Children alive** | | | | | | | | | | | | | | |
| 0 | 50 | 27.3 | 1 | – | 1.00 | – | 1 | – | – | – | – | – | 1 | – |
| 1 | 52 | 28.4 | 0.44 | 0.18 to 1.08 | 0.44 | 0.18 to 1.10 | 0.49 | 0.19 to 1.24 | – | – | – | – | 0.2 | 0.04 to 1.12 |
| 2 | 40 | 21.9 | 0.37 | 0.15 to 0.95 | 0.39 | 0.15 to 1.02 | 0.46 | 0.18 to 1.22 | – | – | – | – | 0.13 | 0.02 to 0.76 |
| 3 | 24 | 13.1 | 0.27 | 0.10 to 0.75 | 0.27 | 0.10 to 0.74 | 0.31 | 0.11 to 0.87 | – | – | – | – | 0.13 | 0.02 to 0.80 |
| 4 | 17 | 9.3 | 0.24 | 0.08 to 0.72 | 0.24 | 0.08 to 0.75 | 0.25 | 0.08 to 0.78 | – | – | – | – | 0.05 | 0.01 to 0.36 |
| **Intermediate determinants** | | | | | | | | | | | | | | |
| **Knowledge about HIV risks** | | | | | | | | | | | | | | |
| Good | 162 | 88.5 | 1.18 | 0.53 to 2.63 | – | – | – | – | – | – | – | – | – | – |
| Poor | 21 | 11.5 | 1 | – | – | – | – | – | – | – | – | – | – | – |
| **Risk perception for HIV infection** | | | | | | | | | | | | | | |
| Own high-risk behaviour | 37 | 20.2 | 1.1 | 0.38 to 3.16 | – | – | – | – | – | – | – | – | – | – |
| Partner(s)' high-risk behaviour | 92 | 50.3 | 1.06 | 0.42 to 2.68 | – | – | – | – | – | – | – | – | – | – |
| Other reasons | 36 | 19.7 | 0.81 | 0.29 to 2.26 | – | – | – | – | – | – | – | – | – | – |
| None | 18 | 9.8 | 1 | – | – | – | – | – | – | – | – | – | – | – |
| **Peer influence** | | | | | | | | | | | | | | |
| Relative(s) on ART | 72 | 39.3 | 9.12 | 4.25 to 19.58 | – | – | – | – | 1.95 | 0.70 to 5.42 | 1.83 | 0.66 to 5.06 | 1.79 | 0.61 to 5.20 |
| Friend(s) on ART | 91 | 49.7 | 16.38 | 7.22 to 37.20 | – | – | – | – | 3.04 | 1.03 to 8.93 | 2.52 | 0.85 to 7.46 | 2.19 | 0.69 to 6.97 |
| None | 20 | 10.9 | 1 | – | – | – | – | – | 1 | – | 1 | – | 1 | – |
| **Sickness in last 12 months** | | | | | | | | | | | | | | |
| HIV-related illness | 69 | 37.7 | 1.09 | 0.57 to 2.09 | – | – | – | – | – | – | – | – | – | – |
| Other illness | 42 | 23.0 | 0.69 | 0.35 to 1.35 | – | – | – | – | – | – | – | – | – | – |
| None | 72 | 39.3 | 1 | – | – | – | – | – | – | – | – | – | – | – |

Continued

**Table 3** Continued

| | Bivariate analysis | | | | Socio-demographic | | Socio-demographic sex work | | Intermediate determinants | | Intermediate determinants sex work | | Full model | |
|---|---|---|---|---|---|---|---|---|---|---|---|---|---|---|
| | n | % | AOR | 95% CI | AOR | 95% CI | AOR | 95% CI | AOR | 95% CI | AOR | 95% CI | AOR | 95% CI |
| STD symptoms in last 12 months | | | | | | | | | | | | | | |
| Yes | 51 | 27.9 | 1.03 | 0.56 to 1.89 | – | – | – | – | – | – | – | – | – | – |
| No | 132 | 72.1 | 1 | – | – | – | – | – | – | – | – | – | – | – |
| Psychological distress | | | | | | | | | | | | | | |
| Yes | 36 | 19.7 | 0.46 | 0.24 to 0.86 | – | – | – | – | 0.43 | 0.19 to 0.99 | 0.41 | 0.18 to 0.96 | 0.48 | 0.20 to 1.18 |
| No | 147 | 80.3 | 1 | – | – | – | – | – | 1 | – | 1 | – | 1 | – |
| Pregnancies in last 3 years | | | | | | | | | | | | | | |
| One or more | 28 | 15.3 | 0.86 | 0.41 to 1.80 | – | – | – | – | – | – | – | – | – | – |
| None | 155 | 84.7 | 1 | – | – | – | – | – | – | – | – | – | – | – |
| Travel time to ART service * | | | | | | | | | | | | | | |
| <30 min | 43 | 23.5 | 1 | – | – | – | – | – | 1 | – | 1 | – | 1 | – |
| 30–59 min | 42 | 23.0 | 0.27 | 0.08 to 0.86 | – | – | – | – | 0.27 | 0.08 to 0.89 | 0.34 | 0.10 to 1.15 | 0.41 | 0.11 to 1.58 |
| 60–89 min | 34 | 18.6 | 0.32 | 0.09 to 1.18 | – | – | – | – | 0.37 | 0.10 to 1.40 | 0.42 | 0.11 to 1.62 | 0.34 | 0.07 to 1.67 |
| 90 min | 63 | 34.4 | 0.37 | 0.12 to 1.16 | – | – | – | – | 0.46 | 0.14 to 1.46 | 0.51 | 0.15 to 1.65 | 0.65 | 0.17 to 2.49 |
| Uncertain | 1 | 0.5 | – | – | – | – | – | – | – | – | – | – | – | – |
| Stigma and discrimination (in the community) | | | | | | | | | | | | | | |
| Yes | 38 | 20.8 | 1.3 | 0.64 to 2.64 | – | – | – | – | – | – | – | – | – | – |
| No | 145 | 79.2 | 1 | – | – | – | – | – | – | – | – | – | – | – |

*Includes women not aware of HIV testing and ART services to prevent exclusion of these participants from the multivariable analysis. ORs were not interpreted for this group as they are not comparable with the reference category.

AOR, adjusted OR; ART, antiretroviral treatment; STD, sexually transmitted diseases.

of infected FSWs who had been diagnosed (58.2%) was slightly higher than estimates for FSWs in urban Zimbabwean locations (50% in 2013)[12]; while the proportion of those diagnosed who had been started on treatment (87% vs 0%–73%) and the proportion of those on treatment who reported adhering to ART (91% vs 67%–100%[10 32 33]) were also high. We are unaware of any previous studies that have compared uptake of HIV testing and ART in representative samples of FSWs and NSWs from the same population. However, similar levels of ART adherence have been found in Mozambique and Benin.[34 35]

The results suggest that several of the structural, interpersonal and personal factors outlined in our framework may contribute to differences in uptake of HIV testing and ART services between and among FSWs and NSWs. As noted previously by Paulin and colleagues in a rural setting in Mozambique,[36] knowledge of HTC services can affect its uptake among women (42%). In Manicaland, FSWs had better knowledge of ART and, because they lived largely in towns, were structurally advantaged over NSWs, who more often lived in areas more remote from testing and treatment facilities. In terms of uptake of HIV care, these factors appear to have offset the disadvantages that FSWs face from poorer mental health and greater stigma and discrimination. Poor mental health, in the form of greater psychological distress, is associated with lower ART uptake in east Zimbabwe.[29] As in many previous studies,[37] we found that psychological distress was more common in sex workers but we did not find this was an important factor mediating uptake. FSWs in this study also reported higher levels of stigma linked to HIV than NSWs; however, unlike studies elsewhere in Zimbabwe,[38] we did not find stigma to be a significant deterrent to accessing healthcare. One reason may be that our study questionnaire did not include a measure of stigma specifically related to sex work. FSWs were more likely to report HIV illness and be HIV positive, but perception of HIV-related symptoms, not HIV serostatus, was associated with HIV testing. This suggests women in Manicaland, and particularly FSWs, are likely to be diagnosed and prescribed treatment late, which can mean reduced survival[39] and greater HIV-related comorbidities. It could be that HIV-positive NSWs may have more pregnancies and therefore often get diagnosed early when still healthy, and HIV-positive FSWs have fewer pregnancies and therefore often only get diagnosed late after becoming sick.

This study uses a unique data source that draws from the combined strengths of population surveys and chain-referral methods and allowed us to analyse a representative sample with reduced under-reporting of locally resident FSWs, and provided a rare opportunity to compare the characteristics and determinants of HIV service use for FSWs and NSWs from the same study areas. However, the data used were cross-sectional so we have been unable to determine the causal nature of the relationships explored in the study. Also, in comparing the HIV care cascades for FSWs and NSWs, we have used self-reported ART adherence as a proxy for viral suppression as biomarkers for viral load were not available.[40] Finally, our study sites were not covered by FSW intervention programmes in Zimbabwe such as 'Sisters with a Voice' or the SAPPHIRE trial (http://www.ceshhar.org.zw/). This has allowed us to compare the experience of FSWs and NSWs in the absence of targeted interventions; however, it would also be valuable for researchers with data from areas where these interventions are being implemented to perform a comparable analysis.

In east Zimbabwe, between 2009 and 2011, FSWs were more likely than NSWs to have been tested for HIV infection and to have taken up ART. These findings challenge the common perception that HIV-infected FSWs are marginalised from HIV treatment in the absence of targeted services. However, high ART coverage in FSWs is critical both for their own health and survival (with many FSWs appearing to access treatment only at advanced stages of infection) and to reduce the rate of new HIV infections in the general population. Furthermore, the results of this study show that different factors influence uptake of HIV services in FSWs compared with NSWs. For example, while decentralised services (including use of recently developed sensitive and specific rapid tests to 'task-shift' HIV testing to community health workers) and intensified efforts to improve personal risk perception and may increase uptake of ART in NSWs, targeted services that address the stigma and discrimination associated with sex work (not measured here but shown to be an important factor in other studies[37 41]) may be more effective for FSWs. NSWs would also additionally benefit from measures to improve treatment uptake once diagnosed. One possible approach for this could be to use couples' HIV testing and counselling to address the dominant interpersonal role of male spouses in determining women's HIV care that has been described in previous qualitative research in Manicaland.(35) The effect of such unmeasured influences could be reflected in the residual effect of sex work after adjusting for socio-demographic factors and intermediate determinants. Further research is needed as the situation may be changing, particularly since the introduction of universal eligibility for ART; nevertheless, recent data show that a third of HIV-infected women in Zimbabwe are not yet virally suppressed(36) so continued and enhanced efforts such as these are probably still needed to increase coverage of treatment services in both FSWs and NSWs.

**Contributors** RR, JE, PJW, SG, CAN and KN were involved in study concept and design. CAN, JE, KN and AT acquired and curated the data. JE, RR, SG and EO were involved in the design of the analysis. RR conducted the statistical analysis supervised by SG and JE. RR, SG, JE and EO interpreted the results and drafted the article.

**Funding** SG thanks the Wellcome Trust for funding (grants: 084401/Z/07/B and 090285MA). JE thanks the Medical Research Council for her PhD funding (grant number http://www.mrc.ac.uk/index.htm) and the Wellcome Trust for postdoctoral funding (grant number: 090285/Z/09/Z; http://www.wellcome.ac.uk/). PJW thanks the MRC for Centre funding (MR/K010174/1) and the UK NIHR Health Protection Research Unit in Modelling Methodology at Imperial College London in partnership with Public Health England for funding (HPRU-2012-10080).

**Disclaimer** The views expressed are those of the authors and not necessarily those of the Department of Health, MRC, NHS, NIHR, Public Health England or the Wellcome Trust.

**Competing interests** None declared.

**Patient consent** Obtained.

**Ethics approval** Prior ethical approval for the Manicaland study (with the WR study included as a substudy) was obtained from the Medical Research Council of Zimbabwe (MRCZ/A/681) and the Imperial College Research Ethics Committee (ICREC_9_3_13).

**Provenance and peer review** Not commissioned; externally peer reviewed.

**Data sharing statement** Data produced by the Manicaland Project can be obtained from the project website: http://www.manicalandhivproject.org/data-access.html. Here we provide a core dataset which contains a sample of socio-demographic, sexual behaviour and HIV testing variables from all six rounds of the main survey, as well as data used in the production of recent academic publications. If further data are required, a data request form must be completed (available to download from our website) and submitted to s.gregson@imperial.ac.uk. If the proposal is approved, we will send a data sharing agreement which must be agreed upon before we release the requested data.

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
