## [Reviewer comments · BMJ Open]

ARTICLE DETAILS

TITLE (PROVISIONAL)	Do Female Sex Workers have Lower Uptake of HIV Treatment Services than Non-Sex-Workers? A Cross-sectional Study from East Zimbabwe
AUTHORS	Rhead, Rebecca; Elmes, Jocelyn; Otobo, Eloghene; Nhongo, Kundai; Takaruzza, Albert; White, Peter; Nyamukapa, Constance; Gregson, Simon

VERSION 1 – REVIEW

REVIEWER	Joseph KB Matovu Makerere University School of Public Health, Kampala, Uganda
REVIEW RETURNED	13-Aug-2017

GENERAL COMMENTS	The paper, “Do female sex workers have lower uptake of HIV treatment services than non-sex-workers? A case study from East Zimbabwe” shares important findings that show that HIV testing and treatment initiation was higher in female sex workers than non-sex-work women, although adherence to treatment was similar between both groups. I have a few comments that the authors should address before the paper is accepted for publication. General comments 1. On page 9, lines 162-3, the authors write, “seeds were selected to represent the diversity of those involved in sex exchange. The seeds then recruited up to three peers that met broad eligibility criteria...” How did the authors ensure that the women recruited by the seeds were indeed sex workers? Given that participants would receive a bar of laundry soap, the possibility that the seeds recruited their friends who might not necessarily have been sex workers is high. Authors should include a description of how they guarded against recruitment of women as sex workers who were not indeed sex workers.2. In the same vein, there is a possibility that non-sex-work women could have included female sex workers, except that these women did not identify themselves as sex workers. How did the authors ensure that non-sex-work women did not include undeclared sex workers?3. In general, the methods section and indeed the results section of the paper dwell more on female sex workers than non-sex-workers. Since the paper compares uptake of HIV services between the two categories of women, it would be important to provide equal treatment to each of the two categories of women.
--

For instance, in the description of variables, only female sex workers are defined. Who were the non-sex-workers? These should be defined in much the same way as the female sex workers. Also, in the results section, while there is a clear description of how the 174 women were identified right from 3,402 women, no systematic description is presented on how the 2,555 non-sex-work women were identified. I suggest that the authors revise this section to show the total number of non-sex-work women that were enrolled in both studies; and then present a systematic breakdown of how this number was reduced to 2,555 eligible women.

4. In the results section, the authors note that of the 174 women identified as female sex, only 31 were identified in both studies. My understanding was that women who were enrolled in the study had to have been identified in both studies (see page 8, line 168), rather than in at least one of the studies. The authors should throw more light on why the number of female sex workers identified in both studies is so small yet the impression created on page 8 suggests that all the women should have participated in both studies.

5. The presentation of results on page 13-15 is heavily bent on presenting the socio-demographic characteristics and intermediate determinants of HIV service use, HIV testing and HIV treatment of FSWs than non-sex-workers. While some limited comparison is made, I would expect that the comparison between the two groups should form the central focus of the paper rather than mentioning it in passing. This aspect is related to my comment in item 3 above.

6. How did the theoretical framework referred to in the Methods section help in the discussion of the study findings?

Specific comments

Abstract

1. The conclusion on the abstract seems to negate the findings presented in the 'methods' section of the abstract. How is this conclusion informed by the theoretical framework that was adopted to guide the implementation of the study? The authors should throw more light on this.

Methods

1. On page 9, the authors define, among others, the 'socio-demographic characteristics' that the authors considered to be potential underlying determinants of women's involvement in sex work and use of services. I suggest that the authors include a citation to qualify the characteristics listed as being potentially associated with the outcome.

2. Line 224, page 10: The word 'aware' should be "awareness", and the word 'participants' should have an apostrophe added – for that matter.

3. Line 225, page 10: should be edited to read, "... of a health facility offering HTC (or ART), and the estimated travel time to the nearest health facility"

4. It seems that the authors used $p < 0.1$ to determine statistical significance, at least based on what's presented on page 11 (lines 243-246). $P < 0.1$ is too restrictive for the final model. Why didn't the authors use a more liberal statistical threshold, e.g. $p < 0.05$ in the final model?

	Results  Page 12, line 267: the authors should add the words, “than NSW” after ‘... had an HIV test’ Line 307, page 14: the word “also” can be dropped without losing meaning Discussion  In lines 332-335 (page 16), the authors write, “FSW’s greater knowledge about HIV and greater personal ...may have contributed to their higher levels of HIV testing”. While this statement may be true if we consider the findings shared in the Tables, it is largely unsupported by the data presented in the results section. For instance, in the results section, nothing is presented about FSW’s knowledge about HIV or their personal perceived risk of being HIV-positive. And, most importantly, the statement is one-sided, alluding to only FSWs, yet the study aimed to present a comparison between FSWs and NSWs. The authors should ensure that the discussion is based on results presented, and that there is a balanced discussion of the findings as they pertain to the two groups of women. When I look at Table 1, I can see that only 6.8% of FSW as opposed to 93.2% of NSW had good knowledge about HIV risks. So, on what results is the assertion that “FSWs had greater knowledge about HIV” based? On page 17, line 362 refers to a discussion on knowledge of HTC services and how this can affect uptake (of---?) among women, and provides a percentage of 42%. In reference to comment#1 in this sub-section, there is no mention of knowledge of HTC in the results section of the paper. What role does the cited percentage (42%) serve in our understanding of the preceding and proceeding statements? Tables Why are the absolute percentages presented ending with a dot? For instance, “4.%”, “92.%” in Table 1. Since all the other numbers carry one decimal place, I suggest that absolute numbers be presented with a ‘zero’ at the end, e.g. 4.0; 92.0. In addition, since the column heading already has the percentage sign (%), it does not help to repeat the same sign on the numbers presented in the different Tables.
--	---

REVIEWER	Mohammad Karamouzian The university of British Columbia, Canada
REVIEW RETURNED	20-Sep-2017

GENERAL COMMENTS	Thank you for the opportunity of reviewing this manuscript. The manuscript compares HIV treatment cascade between female sex workers and 'non-sex-workers' and concludes that FSWs do not have a lower uptake of HIV treatment services than non-sex-workers. While the paper is well written and the authors have tried to follow all the necessary steps to answer their research question, I have some concerns about the rationale of the study and the implications of their findings. I am having a hard time making sense of the comparison made here. Isn't it obvious that given the targeted services catered towards key populations at risk of HIV across various international settings, their HIV-related knowledge, attitude, and practices are relatively higher than women in the general population?
--

	The authors state that they "are unaware of any previous studies that have compared uptake of HIV testing and ART in representative samples of FSW and NSW from the same population". While they think this is due to the novelty of their work, I have a different opinion and think that this could be attributed to the rationale behind this research question. None of the findings reported are novel or surprising to me. I could actually guess what their main findings would look like when I read the Methods. Moreover, the authors do not elaborate on the implications of their findings for policy, research, and practice. I think the "SO WHAT" piece of the paper is missing. Overall, I think this study is not a great contribution to the literature and needs some work before it is ready for publication. In particular, the framing of the research question should be modified or supported with a strong rationale. I hope the authors find this review helpful.
--	---

VERSION 1 – AUTHOR RESPONSE

Reviewer: 1

Reviewer Name: Joseph KB Matovu

Institution and Country: Makerere University School of Public Health, Kampala, Uganda

Please state any competing interests or state 'None declared': None declared

Please leave your comments for the authors below:

The paper, "Do female sex workers have lower uptake of HIV treatment services than non-sex-workers? A case study from East Zimbabwe" shares important findings that show that HIV testing and treatment initiation was higher in female sex workers than non-sex-work women, although adherence to treatment was similar between both groups. I have a few comments that the authors should address before the paper is accepted for publication.

General comments

1. On page 9, lines 162-3, the authors write, "seeds were selected to represent the diversity of those involved in sex exchange. The seeds then recruited up to three peers that met broad eligibility criteria..." How did the authors ensure that the women recruited by the seeds were indeed sex workers? Given that participants would receive a bar of laundry soap, the possibility that the seeds recruited their friends who might not necessarily have been sex workers is high. Authors should include a description of how they guarded against recruitment of women as sex workers who were not indeed sex workers.

Response: Thank you for pointing this out. Clearly, there is virtually no way to completely guard against the recruitment of non-sex workers in any self-referral process. However, we mitigated impersonation and duplication by cross-referencing names of nominated individuals with the names of women appearing to interview and by close monitoring by our key informants (women with personal experience of sex work or who worked closely with women selling sex in the communities). Based on information provided in a formative study, in the questionnaire we asked women if they had ever or recently practiced any of eight different measures of sex exchange.

Based again on the formative study, we restricted the analysis sample to women reporting sex exchanges that were most commonly reported by sex workers namely, self-identifying as sex workers, meeting paying clients in bars or receiving money for exchange of sex were included. We have added in a section to clarify our strategies for mitigating impersonation and duplication.

2. In the same vein, there is a possibility that non-sex-work women could have included female sex workers, except that these women did not identify themselves as sex workers. How did the authors ensure that non-sex-work women did not include undeclared sex workers?

Response: Our sample of female non-sex workers were women who had not participated in the Women at Risk study, and who's responses to the Manicaland survey indicated that they had never been a sex worker or exchanged sex for goods, food or money. As per our response above, it is unfortunately a limitation of all measurement that relies on self-report.

We feel that it is a major strength of this study that we were able to identify some (perhaps most) of the FSW in a general population survey who normally get missed due to social desirability reporting bias. However, it is true that there may have been some other women in the survey who were missed in the WR study and therefore were still not identified as FSW.

3. In general, the methods section and indeed the results section of the paper dwell more on female sex workers than non-sex-workers. Since the paper compares uptake of HIV services between the two categories of women, it would be important to provide equal treatment to each of the two categories of women. For instance, in the description of variables, only female sex workers are defined. Who were the non-sex-workers? These should be defined in much the same way as the female sex workers. Also, in the results section, while there is a clear description of how the 174 women were identified right from 3,402 women, no systematic description is presented on how the 2,555 non-sex-work women were identified. I suggest that the authors revise this section to show the total number of non-sex-work women that were enrolled in both studies; and then present a systematic breakdown of how this number was reduced to 2,555 eligible women.

Response: Thank you for pointing this out to us. We have revised the methods and results section of the paper to include further detail on our non-sex worker sample and how they were selected.

4. In the results section, the authors note that of the 174 women identified as female sex, only 31 were identified in both studies. My understanding was that women who were enrolled in the study had to have been identified in both studies (see page 8, line 168), rather than in at least one of the studies. The authors should throw more light on why the number of female sex workers identified in both studies is so small yet the impression created on page 8 suggests that all the women should have participated in both studies.

Response: All FSW in this analysis participated in the Manicaland survey but only a subset participated in the WR study. Thirty-two FSW did not participate in the WR study but indicated in the Manicaland survey that they had engaged in sex work or exchanged sex. 142 FSW had participated in both surveys and indicated that they had engaged in sex work in the WR study (n=111) or both studies (n=31).

5. The presentation of results on page 13-15 is heavily bent on presenting the socio-demographic characteristics and intermediate determinants of HIV service use, HIV testing and HIV treatment of FSWs than non-sex-workers. While some limited comparison is made, I would expect that the comparison between the two groups should form the central focus of the paper rather than mentioning it in passing. This aspect is related to my comment in item 3 above.

Response: This is an excellent point. However, the emphasis of the paper is on FSWs as the main novelty of the study lies in its ability to compare their experiences with those of a representative sample of women in the general population within the same geographical locations. We have edited the text throughout the paper to emphasise that numerous comparisons of FSW vs NSW that are made.

6. How did the theoretical framework referred to in the Methods section help in the discussion of the study findings?

Response: The theoretical framework referred to in the methods section was developed to guide our analysis of the roles that involvement in sex work and its consequences can play in mediating associations between underlying socio-demographic characteristics and use of HIV testing and treatment services. The findings presented in our results section show the associations between sex work and the socio-demographic and intermediate determinants outlined in our framework. These findings are then summarised and discussed in our discussion section. We have added into our discussion where findings agree and refute the framework and discuss at length how the structural, interpersonal and personal factors highlighted in the framework may contribute to differences in uptake.

Specific comments:

Abstract

1. The conclusion on the abstract seems to negate the findings presented in the 'methods' section of the abstract. How is this conclusion informed by the theoretical framework that was adopted to guide the implementation of the study? The authors should throw more light on this.

Response: We have amended the conclusion portion of our abstract to report the conclusions informed by our theoretical framework.

Methods

1. On page 9, the authors define, among others, the 'socio-demographic characteristics' that the authors considered to be potential underlying determinants of women's involvement in sex work and use of services. I suggest that the authors include a citation to qualify the characteristics listed as being potentially associated with the outcome.

Response: Thank you for pointing this out. We have included a number of citations and refer the reader to the Supplementary material which discusses these items in greater depth.

2. Line 224, page 10: The word 'aware' should be "awareness", and the word 'participants' should have an apostrophe added – for that matter.

Response: This change has been made.

3. Line 225, page 10: should be edited to read, "... of a health facility offering HTC (or ART), and the estimated travel time to the nearest health facility"

Response: This change has been made.

4. It seems that the authors used $p < 0.1$ to determine statistical significance, at least based on what's presented on page 11 (lines 243-246). $P < 0.1$ is too restrictive for the final model. Why didn't the authors use a more liberal statistical threshold, e.g. $p < 0.05$ in the final model?

Response: The point about using a less restrictive determinant of statistical significance is valid but $p < 0.1$ is less restrictive (i.e. more liberal) than $p < 0.05$ so on this basis $p < 0.1$ is reasonable.

Results

1. Page 12, line 267: the authors should add the words, "than NSW" after '... had an HIV test'

Response: This change has been made.

2. Line 307, page 14: the word "also" can be dropped without losing meaning

Response: This word has been removed.

Discussion

1. In lines 332-335 (page 16), the authors write, "FSW's greater knowledge about HIV and greater personal ... may have contributed to their higher levels of HIV testing". While this statement may be true if we consider the findings shared in the Tables, it is largely unsupported by the data presented in the results section. For instance, in the results section, nothing is presented about FSW's knowledge about HIV or their personal perceived risk of being HIV-positive. And, most importantly, the statement is one-sided, alluding to only FSWs, yet the study aimed to present a comparison between FSWs and NSWs. The authors should ensure that the discussion is based on results presented, and that there is a balanced discussion of the findings as they pertain to the two groups of women.

Response: These findings are reported and compared for FSW versus NSW in Table 1, as the reviewer points out in the next comment.

2. When I look at Table 1, I can see that only 6.8% of FSW as opposed to 93.2% of NSW had good knowledge about HIV risks. So, on what results is the assertion that "FSWs had greater knowledge about HIV" based?

Response: This statement by the reviewer contradicts their previous point that the authors do not present findings on FSW's knowledge about HIV or their personal perceived risk of being HIV-positive. Indeed, as the reviewer has pointed out in this comment, such findings are displayed in Table 1.

We believe that the proportions as they were displayed in Table 1 may have been confusing, and have displayed the within group proportions which are easier to interpret. The updated Table 1 now illustrated that 55% ($n=96$) of FSW vs 86% ($n=2210$) of NSW perceive no risk of infection, and by contrast, 22% ($n=39$) of FSW and 1% ($n=41$) of NSW perceive risk from their own high-risk behaviour.

3. On page 17, line 362 refers to a discussion on knowledge of HTC services and how this can affect uptake (of---?) among women, and provides a percentage of 42%. In reference to comment#1 in this sub-section, there is no mention of knowledge of HTC in the results section of the paper. What role does the cited percentage (42%) serve in our understanding of the preceding and proceeding statements?

Response: The line referenced here has been edited to improve clarity.

Tables

1. Why are the absolute percentages presented ending with a dot? For instance, "4.%" , "92.%" in Table 1. Since all the other numbers carry one decimal place, I suggest that absolute numbers be presented with a 'zero' at the end, e.g. 4.0; 92.0. In addition, since the column heading already has the percentage sign (%), it does not help to repeat the same sign on the numbers presented in the different Tables.

Response: Absolute percentages presented in our tables no longer end with a dot. They are presented with a zero as suggested.

Reviewer: 2

Reviewer Name: Mohammad Karamouzian

Institution and Country: The university of British Columbia, Canada

Please state any competing interests or state 'None declared': None

Please leave your comments for the authors below

Thank you for the opportunity of reviewing this manuscript. The manuscript compares HIV treatment cascade between female sex workers and 'non-sex-workers' and concludes that FSWs do not have a lower uptake of HIV treatment services than non-sex-workers.

Comment: While the paper is well written and the authors have tried to follow all the necessary steps to answer their research question, I have some concerns about the rationale of the study and the implications of their findings. I am having a hard time making sense of the comparison made here. Isn't it obvious that given the targeted services catered towards key populations at risk of HIV across various international settings, their HIV-related knowledge, attitude, and practices are relatively higher than women in the general population?

The authors state that they "are unaware of any previous studies that have compared uptake of HIV testing and ART in representative samples of FSW and NSW from the same population". While they think this is due to the novelty of their work, I have a different opinion and think that this could be attributed to the rationale behind this research question. None of the findings reported are novel or surprising to me. I could actually guess what their main findings would look like when I read the Methods.

Moreover, the authors do not elaborate on the implications of their findings for policy, research, and practice. I think the "SO WHAT" piece of the paper is missing. Overall, I think this study is not a great contribution to the literature and needs some work before it is ready for publication. In particular, the framing of the research question should be modified or supported with a strong rationale. I hope the authors find this review helpful.

Response: Thank you for reviewing our paper and providing us with feedback. We would like to correct a possible misunderstanding that this study explores uptake "targeted services catered towards key populations at risk of HIV". This study actually explores uptake of all HIV services in the area, not sex worker targeted services, and at a time when there were no extra efforts to provide targeted services to FSWs (as described in the discussion); this form of comparison has very rarely been done. It is therefore not at all obvious or generally accepted that FSW have better uptake of generalised ART services as evidenced by the references already cited in the paper.

Finally, we would also like to point out that in contrast to reviewer 2's comment that none of the findings are novel, a 2016 review of the care cascade in key populations found the care continuum among FSW has not been well-characterised (Risher, 2016) and in a systematic review of ART uptake by FSW globally only 1 study was included from Southern Africa (Mountain, 2014).

VERSION 2 – REVIEW

REVIEWER	Joseph KB Matovu Makerere University School of Public Health
REVIEW RETURNED	17-Nov-2017
GENERAL COMMENTS	I don't have any additional comments